# Climate Change and Health in Urban Areas with a Mediterranean Climate: A Conceptual Framework with a Social and Climate Justice Approach

**DOI:** 10.3390/ijerph191912764

**Published:** 2022-10-06

**Authors:** Marc Marí-Dell’Olmo, Laura Oliveras, Lourdes Estefanía Barón-Miras, Carme Borrell, Tomás Montalvo, Carles Ariza, Irma Ventayol, Lilas Mercuriali, Mary Sheehan, Anna Gómez-Gutiérrez, Joan Ramon Villalbí

**Affiliations:** 1Agència de Salut Pública de Barcelona (ASPB), Pl. Lesseps 1, 08023 Barcelona, Spain; 2Institut d’Investigació Biomèdica Sant Pau (IIB SANT PAU), Sant Quintí 77-79, 08041 Barcelona, Spain; 3Centro de Investigación Biomédica en Red de Epidemiología y Salud Pública (CIBERESP), Av. Monforte de Lemos, 3-5, 28029 Madrid, Spain; 4Department of Preventive Medicine and Epidemiology, Hospital Clinic, Universitat de Barcelona, Villarroel 170, 08036 Barcelona, Spain; 5Department of Experimental and Health Sciences, Universitat Pompeu Fabra, Doctor Aiguader 88, 08003 Barcelona, Spain; 6Oficina de Canvi Climàtic i Sostenibilitat, Ajuntament de Barcelona, Av. Diagonal 240, 08018 Barcelona, Spain; 7Johns Hopkins Bloomberg School of Public Health, Department of Health Policy and Management, 615 N. Wolfe Street, Baltimore, MD 21205, USA; 8Joint Johns Hopkins University-Pompeu Fabra University Public Policy Center, Universitat Pompeu Fabra, Ramon Trias Fargas, 25-27, 08005 Barcelona, Spain

**Keywords:** climate change, Mediterranean climate, urban, health, health inequalities, public health, conceptual framework

## Abstract

The consequences of climate change are becoming increasingly evident and highlight the important interdependence between the well-being of people and ecosystems. Although climate change is a global phenomenon, its causes and consequences vary dramatically across territories and population groups. Among settings particularly susceptible to health impacts from climate change are cities with a Mediterranean climate. Here, impacts will put additional pressure on already-stressed ecosystems and vulnerable economies and societies, increasing health inequalities. Therefore, this article presents and discusses a conceptual framework for understanding the complex relationship between climate change and health in the context of cities with Mediterranean climate from a social and climate justice approach. The different elements that integrate the conceptual framework are: (1) the determinants of climate change; (2) its environmental and social consequences; (3) its direct and indirect impacts on health; and (4) the role of mitigation and adaptation policies. The model places special emphasis on the associated social and health inequalities through (1) the recognition of the role of systems of privilege and oppression; (2) the distinction between structural and intermediate determinants of climate change at the root of health inequalities; (3) the role of individual and collective vulnerability in mediating the effects of climate change on health; and (4) the need to act from a climate justice perspective to reverse health inequities.

## 1. Introduction

The consequences of climate change are becoming increasingly evident and highlight the important interdependence between the well-being of people and ecosystems. Climate change is considered the greatest health threat of this century, although it is difficult to precisely estimate the magnitude and timing of some of its health impacts. These will depend, in part, on the major effort towards decarbonisation required to achieve the goals of international treaties such as the 2015 Paris Agreement. However, even with such major effort, adaptation to a warmer, more variable climate will be needed. Consequently, a growing number of national, regional and local governments have declared a “climate emergency”; this formal step has prompted commitment to develop action plans tackling the causes of climate change, and incorporating measures to accelerate adaptation to extreme weather events to reduce their short- and long-term effects [1,2].

In this context, public health services are of great importance [2,3]. Many of their essential functions, such as research, epidemiological surveillance and health promotion and protection are essential for climate change mitigation and adaptation. In recent decades, public health services have already responded in specific ways to the different challenges posed by a changing climate. Examples include the 2003 heat wave in Europe that caused more than 50,000 excess deaths [4,5,6,7] or the growth of the mosquito population which has led to the emergence or re-emergence of vector-borne diseases such as malaria and various arboviruses (West Nile virus, dengue, Zika and chikungunya) [8]. The ever-clearer link between these and other phenomena and climate change has given rise to the search for a more integrated approach. A more intersectoral response is therefore needed, with adaptation responses rooted in the local territory and mitigation policies that go beyond the local framework. Here too, public health has a key role to play in ensuring that the design and implementation of mitigation and adaptation policies include social and health co-benefits [9].

Different conceptual frameworks have been developed in an attempt to understand the complex relationship between climate change and health [9,10,11,12,13,14,15]. However, although climate change is a global phenomenon, its determinants, its environmental and social consequences, and its impacts on the health and well-being of populations vary dramatically. For this reason, no framework can be applied to all possible social, economic, environmental and political settings [11].

Among settings particularly susceptible to health impacts from climate change are cities with a Mediterranean climate, which include cities in the Mediterranean Basin, the western United States and Mexico, central Chile, the Cape region of South Africa, and south and southwestern Australia [16]. These cities are located in biogeographical hotspots experiencing some of the strongest effects of climate change [17]. The Mediterranean region, for example, is already warming 20% faster than the global average [18]. Also noteworthy is the increasing ecological drought in these regions due to the expected increased evapotranspiration and reduced rainfall [17]. Several cities with Mediterranean climate are located on the coast and surrounded by mountainous perimeters. This exposes them both to risks related to sea level rise, such as flooding, erosion, and salinization of river deltas and aquifers, and to risks associated with increased wildfires [19,20,21]. Furthermore, the rich Mediterranean biodiversity is at stake, placing complex ecosystems relationships in jeopardy. This could result in the rising of zoonotic, food and vector borne diseases, also enhanced by the projected increment in temperatures [10].

Today, the majority of the population lives in urban areas and this trend is projected to increase [22]. Cities with mild Mediterranean climate and close to the sea are particularly attractive, and therefore are characterised by a high population density and growth. These cities also have a high proportion of older people, who are more vulnerable to the health effects of climate change. Cities are also especially vulnerable to climate change due to their architectural and urban design. Many cities with Mediterranean climates are composed of an ancient and compact network of buildings with extremely low energy efficiency, a high proportion of heat-retaining materials and a lack of green spaces; this intensifies the heat island effect and makes it difficult to cope with extreme temperatures [15,17]. Urban areas also concentrate the highest number of infrastructures and services that could be affected by extreme climate events, often in cascading fashion [23], impacting a large percentage of the population. In the case of cities with a Mediterranean climate, tourism is an essential economic activity. Millions of tourists circulate each year, contributing to all the complex systems that fuel climate change and hindering mitigation and adaptation policies. During the long hot seasons in summer, both locals and visitors tend to spend long hours in outdoor settings, increasing their exposure to environmental factors [24].

Lastly, socioeconomic inequalities in health are usually more marked in urban areas [25]. Climate change may act as an amplifier of these inequalities, as its impacts will put additional pressure on the vulnerable economies and societies that characterise many cities with Mediterranean climates [18]. Therefore, an approach is needed that not only identifies which populations are most vulnerable to the health effects of climate change, but also facilitates an evaluation of where injustices emerge, which affected groups in society are ignored and what processes exist to rectify these injustices.

For all these reasons, this article proposes and discusses a conceptual framework for understanding the complex relationship between climate change and health in the context of cities with Mediterranean climate and from a social and climate justice approach.

## 2. Methods

The conceptual framework described in this paper was built on the experience and knowledge of experts in climate change, public health, and health inequalities and on the available scientific evidence. In 2019, within the framework of the Fall Institute of the Johns Hopkins Bloomberg School of Public Health, we organized the workshop “Planning for climate change” in the city of Barcelona [26,27]. The workshop brought together dozens of experts from different fields and aimed to identify the elements of a public health surveillance system to monitor climate change, its determinants and its effects on health. Through plenary sessions and discussions in small multidisciplinary groups, a preliminary list of topics and indicators was outlined. The workshop also identified the need to: (1) develop a shared conceptual framework adapted to the specificities of cities with a Mediterranean climate, in order to build the monitoring system and guide design and evaluation of health-focused mitigation and adaptation policies; and (2) include equity and justice as key considerations.

Following this first meeting, a working group was set up to conduct a literature review to identify existing conceptual frameworks on climate change and health, and other literature to further contextualise this complex relationship in the setting of cities with a Mediterranean climate as well as within an equity and justice framework. We then went through an iterative process to select, organize, and relate the available information into a conceptual framework. As a starting point, we relied mainly on conceptual models of social determinants of health such as that of the World Health Organisation [28] and that proposed by the Commission for the Reduction of Social Inequalities in Health in Spain [29], as well as previous work on climate change and health by the Lancet Commission [10] and by McMichael [9].

## 3. Description and Discussion of the Conceptual Framework

This conceptual framework focuses exclusively on the anthropogenic causes of climate change and considers both the effects that are already happening and those situations that will probably be faced in the future. Emphasis is placed on the need to combine mitigation and adaptation policies to address climate change and its impacts on health in the long term, but also to provide an immediate response to the difficulties that the climate emergency is already posing, especially for the most vulnerable groups. By focusing on cities with a Mediterranean climate, and mainly middle- and high-income cities, it may miss aspects that may be relevant in other contexts.

This section describes the different elements that integrate the conceptual framework (Figure 1). In short, the model includes: (1) the determinants of climate change; (2) its environmental and social consequences; (3) its direct and indirect impacts on health; and (4) the role of mitigation and adaptation policies. The model places special emphasis on the associated social and health inequalities through: (1) the recognition of the role of systems of privilege and oppression; (2) the distinction between structural and intermediate determinants of climate change and their identification as also being at the root of health inequalities; (3) the role of individual and collective vulnerability in mediating the health effects of climate change on health; and (4) the need to always act from a climate justice perspective to reverse health inequities.

### 3.1. Systems of Privilege and Oppression

Societal systems of privilege and oppression such as capitalism, patriarchy, colonialism, and racism are key roots of both climate change and social inequalities in health. These inequitable systems, based on a set of ideas, beliefs and practices conceived by people, make up the dominant belief system and configure society [30]. The world is thus understood in binary terms with polarities such as superior/inferior (human/nature, North/South, man/woman, etc.) and normal/deviant, which manifest in a hierarchical social order with unequal power relations. Through explicit and non-explicit rules and laws, these unequal power relations allow and reproduce the exclusion and domination of certain social groups and the exploitation and degradation of nature, which leads to social and material benefits for other social groups [30]. By including these systems of privilege and oppression at the background of this conceptual framework, we aim to highlight their persistent embeddedness throughout the process: from the more structural causes of climate change to the vulnerabilities that modify the health impacts of climate change. We want to broaden the focus and make visible not only the social and health inequalities related to climate change, but also the structural processes that generate systematically social and climate injustice [31].

### 3.2. Structural Determinants

Structural determinants consist of the interaction of the socio-economic and political context and axes of inequality. These are strongly conditioned by systems of privilege and oppression and are responsible for the ways of life (intermediate determinants) that drive climate change and social inequalities.

#### 3.2.1. Socioeconomic and Political Context

The context is broadly defined as including all the socioeconomic and political mechanisms facilitating the emission of greenhouse gases (GHG) causing climate change, as well as the creation, configuration and perpetuation of the social stratification and distribution of power and resources within a society [28,29]. This includes, on the one hand, the political tradition and governments, ranging from the local government to the international level, where there are also international bodies that play an important role in halting climate change such as the International Panel on Climate Change (IPCC). On the other hand are the various economic stakeholders, such as the large corporations and social stakeholders such as employers’ associations and non-governmental organizations (NGOs). Together with the culture, values and the sociohistorical legacy of our society, these stakeholders compose the power structures that influences factors important in GHG production, such as: (1) national and international public policies; (2) economic growth and the globalized economy; (3) demographic growth and urbanization; and (4) the overharvesting of natural resources and environmental degradation. Thus, for example, public policies related to the environment, transport, energy, housing, industry, agriculture, and commerce have a huge effect on GHG emissions, as well as on the material and social resources that determine people’s chances of enjoying good health. The current economic model, based on continuous growth with increasing consumption of natural resources, not only threatens the ecological equilibrium of the planet but has also multiplied social inequalities [32]. An example that applies especially to cities with a Mediterranean climate is the growing cruise tourism industry. Cruise ships constitute one of the most energy intense forms of touristic activities [33]. The Mediterranean is the second largest cruise destination and hosts 10% of the global cruise traffic. Between 1990 and 2020, the annual growth rate of passengers in Mediterranean ports was 6.63%, and the trend is upwards. In 2019, 31 million cruise passenger movements were recorded, with an increase of 11.5% compared to 2018 [34]. This activity negatively impacts communities through air and water pollution, economic leakage, poor working conditions and wages, tax avoidance, and overtourism [35,36,37], which worsens the density and over-consumption typical of cities. Although cities represent only 2% of the earth’s surface, they contain 55% of the population, consume 78% of global energy and produce 60% of GHG [38]. In addition, inequalities in health are usually more marked in urban areas [25]. Overharvesting of natural resources reduces the ability of ecosystems to halt environmental degradation and provokes forced displacements and socio-environmental conflicts. For example, due to high vulnerabilities and limited adaptive capacities, North African countries face increased probability of climate-conflict linkages though agricultural production and the food–water–energy nexus [39]. Finally, an important role is played by citizen mobilization, which, through social movements and other forms of citizen participation, can act as an agent of change in the dominant system of social, economic, and political organization. Several social movements have already contributed to climate change mitigation through a wide variety of strategies ranging from awareness-raising to changes in regulation, legislation and policy [40]. For example, changes in California’s climate policy limiting permits for new oil wells are the result of social movements towards a low-carbon society [41].

#### 3.2.2. Axes of Inequality

The various axes of inequality, such as social class, gender, age, ethnicity and geographical origin or place of residence, among others, as well as their intersections, stratify society, creating hierarchies of power between different social groups. These unequal power relations have repercussions on modes of life of different social groups which are related to: (1) higher or lower GHG emissions; and (2) greater or lesser access to social and material resources that affect a person’s chances of enjoying good health and ability to adapt to climate change and avoid its health impacts. For example, people with more economic resources have a larger ecological footprint due to, among other things, greater use of air and private land transport or greater consumerism and waste generation [42,43]. At the same time, however, it is the people with the least social and economic resources who are most exposed to the effects of climate change and who face the greatest difficulties in adapting to and recovering from these effects. For example, Mediterranean cities are characterised by old housing stock and low energy efficiency. In Barcelona, people from the most disadvantaged social classes have been found to live in dwellings with poorer housing conditions and experience more energy poverty [44]. This increases their exposure to rising temperatures and heat waves and reduces their capacity to adapt. Contextual factors of the neighbourhood of residence such as urban planning also have an impact on people’s health. In Haifa, the greatest exposure to heat and heat waves is downtown because it is hotter due to the topography, but also because it is home to the most deprived neighbourhoods, where there are fewer trees, the heat island effect is greater, and there is more energy poverty [45]. Adaptive interventions through urban greening in compact cities such as those in the Mediterranean are essential, but they must be prevented from leading to further gentrification and creating new dynamics of exclusion [46].

It should be noted that the different axes of inequality do not exist in isolation, but that people are crossed by all of them, creating different social realities [47]. These axes interact with individual vulnerabilities that also affect intermediate determinants and health impacts. For example, in cities with a Mediterranean climate, there is a high percentage of elderly people. This is especially the case in northern Mediterranean cities, such as those in France, Spain, Italy and Greece, countries of the European Union with the highest levels of life expectancy [48]. The shares of women aged 65 in these countries are 23%, 23%, 26% and 25%, respectively, and in the case of men 19%, 18%, 21% and 20%, respectively [49]. Among the elderly population, there are more frail people; the biological effect of age interacts with the social effect, defining what it means to be elderly in our society, including lower participation in society and possible discrimination (ageism). Moreover, in this age group, women outnumber men. Older women are more susceptible to heat due to poorer age-related thermoregulation, while at the same time having fewer economic and social resources to protect themselves from heat due to their gender and age.

These axes are key to understanding the concept of climate justice.

### 3.3. Intermediate Determinants

The main causes of GHG emissions are current models of energy, mobility and infrastructure, agri-food, production and reproduction, consumption and waste, urban planning, housing, and sociocultural factors. At the same time, these models modulate how people access the various material and social resources, acting as intermediate determinants of health inequalities [29].

#### 3.3.1. Greenhouse Gases

Atmospheric gases of anthropogenic origin contributing to the greenhouse effect include carbon dioxide (CO_2_), methane (CH_4_), nitrous oxide (N_2_O), ozone, and halocarbons. The concentrations of these GHGs have substantially increased over the last few decades [38]. The framework described here identifies the main models causing anthropogenic GHG emissions. Currently, the main sources of GHG are energy production (due to CO_2_ emissions derived from the carbon-burning process, especially of coal and oil), and transport (due to CO_2_ emissions from motor vehicles). However, also important are housing and industry (especially due to CO_2_ emissions from fuel combustion, which is highly influenced by its energy efficiency), as well as food, due to the globalized production model, which increases emissions through transport of products, displacing traditional consumption of locally-grown food, deforestation of some areas to increase production, and higher meat consumption in the diet, especially from bovines (ruminants produce high levels of methane, an important GHG) [50,51].

The Mediterranean basin, home to 7.4% of the world’s population, is responsible for approximately 6% of global GHG emissions [21]. Current, but also past and expected future emissions, are not homogeneous between the north, south and east of the basin. For example, although historically northern countries have consumed more energy, moderate population growth, increased efficiency and a more stable economy have allowed a decrease in energy demand of 8% since 2010. It is also expected that consumption will continue to decrease in these countries and that the share of renewable energies will continue to increase. In contrast, southern Mediterranean countries have undergone sustained economic and population growth over recent decades and energy demand is expected to continue to increase in the coming decades [21].

#### 3.3.2. Material and Social Resources

The different GHG-generating models mentioned above also define access to material and social resources, which have a direct effect on health. These include employment and working conditions, housework and caregiving, income and the economic situation, housing and the material situation, the residential environment, social capital, social support, and social networks, as well as education. Unequal access to these material and social resources along different axes of inequality, together with lack of power, are the main causes of existing health inequalities [28,29]. At the same time, this unequal access modulates exposure to the effects of climate change and adaptive capacity, thus exacerbating social inequalities in health.

### 3.4. Climate and Consequences

#### 3.4.1. Direct Effects

Higher GHG concentrations cause more heat retention in the atmosphere, leading to a series of environmental phenomena. One of the most important is the increase in the earth’s surface temperature. In a scenario of insufficient policy action to turn around GHG emissions and in which CO_2_ emissions almost double by 2050 (Shared Socio-Economic Pathway SSP3-7.0), it is very likely that the mean temperature of the earth’s surface at the end of the 21st century (2081–2100) will be about 3.6 °C higher, compared to the period 1850–1900 [52]. It is also highly probable that there will be an increase in several extreme phenomena that may vary depending on the location, such as: (1) frequent high daytime and night-time temperatures; (2) frequency and/or duration of heat waves; (3) a decrease in precipitation; (4) the intensity and duration of droughts; (5) storms, torrential rain, and floods; (6) a rise in sea level; and (7) more frequent wildfires.

#### 3.4.2. Indirect Effects

##### Environmental Consequences

The above-mentioned environmental phenomena produced by climate change can trigger other ecological and environmental changes.

Indeed, climate change and globalization are considered to be the main factors explaining the emergence and/or re-emergence of reservoirs, vectors and pathogens worldwide [53]. This phenomenon affects changes in the geographical distribution, seasonality, processes of pathogen transmission and the size of the population of autochthonous vectors, as well as the appearance of new vectors [54,55]. For example, the presence of Aedes albopictus in southern Europe has provided opportunities for the introduction of Aedes mosquito-borne viruses via viraemic humans travelling from endemic regions. Outbreaks of dengue virus have been detected in the city of Nimes in southern France [56], and of chikungunya virus in Italy, notably in the cities of Castiglione di Cervia and Castiglione di Ravenna [57,58]. The best example of the emergence of viruses that have encouraged autochthonous mosquito-borne virus transmission in Europe is that of the West Nile virus, predominantly transmitted by mosquito species of the genus Culex [59]. Alteration of the complex and rich Mediterranean ecosystems may also contribute to increased vector circulation. Studies have reported the association of low bird diversity and increased human risk transmission of West Nile fever [60].

Similarly, climate change can also modify the geographical area, seasonality, survival and growth capacity of several food- and waterborne pathogens, such as *Salmonella* spp., *Escherichia coli*, *Campylobacter* spp. and *Legionella* spp., as studies have reported in the cities of the Mediterranean Middle East during 2021 [61]. The availability of food can also be affected by crop damage caused by extreme climate events and yield losses especially in maize and wheat. Mediterranean countries are net importers of cereal and fodder/feeding products. They will therefore be strongly affected by disruptions in global agricultural markets related to climate change effects located in other producing regions. The lack of water will also affect foods typical of the Mediterranean diet, such as tomatoes (a highly water-demanding crop), and olives, currently a rain-fed crop that could become unviable without irrigation [21]. Ecosystem disruption, both through the migration and extinction of native species and the introduction of invasive alien species, can also lead to the loss of classic crops.

Additionally, processes such as oceanic physical and chemical modifications and disruption of biodiversity can affect future food resources. In the case of the Mediterranean Sea, increases in water temperature, higher than in the oceans, and acidification are expected. This may lead to the local extinction of more than 20% of exploited fish and marine invertebrates by 2050 and proliferation of species with a greater affinity for this environment, such as jellyfish. This would result in loss of marine food species, directly affecting the Mediterranean diet, as well as making coastal waters more dangerous for recreational uses and hindering a resource to cope with heat waves [62,63]. Between 1994 and 2017, total landings from Mediterranean fisheries have already decreased by 28% [21].

Likewise, in periods of drought, the availability and quality of drinking water in areas with a Mediterranean climate can also be compromised. In these areas, it is often necessary to resort to surface water to meet the needs of urban concentrations. In addition, the high population growth due to tourism in many cities with a Mediterranean climate, especially in summer, can further stress water systems during periods of drought. Moreover, extreme precipitation events can lead to floods that compromise drinking water infrastructure, jeopardize water quality and the cleanliness of recreational water (such as beaches, for example). Disastrous flash floods are already frequent and will likely become more frequent in many Mediterranean countries including Italy, France and Spain, affecting coastal areas, in particular, where population and urban settlements are growing in flood-prone areas [21].

Finally, phenomena such as there being fewer days of rainfall, higher temperatures, and wildfires near cities can also influence air quality, provoking an increase in exposure to outdoor air contaminants and the production of secondary contaminants such as ozone. Moreover, these phenomena can lead to higher pollen counts and longer pollen seasons, increasing its allergenicity [64]. Air quality is also affected by two more phenomena. On the one hand, most sources of GHG emissions also emit air contaminants, not only NO_2_, but also inhalable particulate matter (PM). Many cities with Mediterranean climates, such as Barcelona or Rome, already have high levels of air pollution, mainly due to road traffic [65]. On the other hand, in the Mediterranean basin, some areas intermittently receive intrusions of Saharan dust, which in periods of anticyclones and without rainfall can markedly increase the PM pollution already generated by human activity [66].

##### Social Consequences

Climate change, both directly, through temperature increases or extreme weather events, and indirectly, through the above-mentioned environmental effects, can have major socioeconomic consequences. On the one hand, the scarcity of some products and greater demand for them can increase the price of basic goods such as water, food, and energy. Basic sectors such as housing can also be affected, due to damage and changes in the needs of buildings and other infrastructures, or work. The increase in thermal stress is expected to have a strong impact on productivity and occupational risks, which will translate into major job losses and worsening of labour conditions. Although the most affected sectors are expected to be agriculture and construction, other sectors will also be affected, such as transport, tourism, sports and some industrial jobs or even office work, due to the psychological fatigue produced by very high temperatures [67]. In a recent review on the impact of extreme heat on occupational injuries, hot Mediterranean climates were one of the climate zones identified as being at greatest risk [68]. In addition, these areas may experience significant job losses linked to the possible disruption of tourism associated with the consequences of climate change [69,70].

The adverse effects of climate change on living and working conditions will be disproportionately experienced by the most disadvantaged groups, intensifying current social inequalities. On a more global scale, the scarcity of resources and the environmental effects of climate change may provoke an increase in socioeconomic conflicts and migration, as well as increasing the risk of armed conflict. Altogether, climate change may affect social cohesion and community support networks. As an example, several studies have already reported migration fluxes from sub-Saharan countries to the Middle Eastern and North African Mediterranean and related health concerns for both local and migrant populations. It is estimated that 10–20% of this migration is due to extreme weather events [71,72].

### 3.5. Impacts on Health and Health Inequalities

Climate change affects the population’s health in various ways and is considered the greatest health threat of the current century [73,74]. The various impacts of climate change on health can be classified into two categories [3,9]: direct and indirect.

#### 3.5.1. Direct Impacts

Direct effects include the consequences of exposure to extreme climate events, such as extreme temperatures, heatwaves, storms, and floods.

Extreme heat temperature has a direct effect on health by reducing the body’s thermoregulatory ability. The loss of control of internal temperature is related to a series of temperature-related illnesses such as heat syncope, heat exhaustion, heat cramps, and heat stroke [75]. Similarly, extreme temperature can also cause or aggravate cardiovascular, respiratory, and renal (including renal insufficiency) diseases, and prolonged exposure to high temperatures is associated with more hospital admissions for these causes. Extreme heat can also cause or worsen cerebrovascular and gastrointestinal (mainly in children) diseases, diabetes, mental, behavioural, and cognitive disorders and provoke premature birth, and can even lead to premature death from all the above-mentioned causes. Although cold is currently the main cause of mortality attributable to suboptimal temperatures in all countries studied, some studies already indicate that without further efforts to reverse the progress of climate change, in warmer regions such as those with a Mediterranean climate this pattern will be reversed, and heat will become the main cause of mortality attributable to suboptimal temperatures [76]. In many Mediterranean cities, heat is already responsible for a large number of deaths. For example, in Barcelona, it was estimated that for the period 1992–2015, extreme temperatures resulted in more than 3500 deaths and that the risk of mortality increased by 14% in women and 4% in men when temperatures were high [77].

Exposure to other extreme weather events, such as storms or floods, can have direct effects such as injuries and deaths, although these effects are currently less frequent in urban environments with a Mediterranean climate. These events can also produce post-traumatic stress, acute stress, depression, anxiety and other mental health problems, often leading to more than one problem at a time [78]. These mental health problems can, moreover, increase medication use, intake of other substances, such as alcohol, and the number of suicides [79]. For example, in several Mediterranean climate areas, including Italy, Greece and California, an increase in psychiatric consultations and suicides was observed in relation to high extreme temperatures [80,81,82]. Social cohesion and social support networks can also be affected, impairing mental health and wellbeing [11,83]. Finally, extreme climate events can increase respiratory problems (for example, due to exposure to mould in flood-damaged buildings) and diarrhoea.

#### 3.5.2. Indirect Impacts

The environmental and social effects of climate change have numerous indirect effects on health, including malnutrition and an increase in communicable and non-communicable diseases and impaired quality of life, and can even prove fatal. Higher temperatures can increase the incidence of food- and waterborne diseases through several mechanisms, such as encouraging better growth conditions for pathogenic microorganisms (for example, higher temperatures favour some bacteria causing gastrointestinal diseases, such as *Salmonella* spp. and *Campylobacter* spp.) and a change in patterns of exposure to sources of infection. For example, the increase in recreational use of water to cool down may encourage exposure to waterborne pathogens [84,85,86]. Likewise, changes in the pattern of precipitation and temperature can encourage the growth and expansion of arthropod vectors, increasing risk of diseases such as malaria, dengue, chikungunya and Zika; and can increase risk of the re-emergence of diseases such as the plague and hantavirus transmitted by reservoirs such as murids (mice and rats) [54,87,88]. Moreover, the expected effect of climate change on human migration could increase the number of imported cases of diseases that are non-endemic in our environment and that, in favourable conditions, could initiate cycles of autochthonous transmission.

Changes in air quality could influence the incidence of cardiovascular and respiratory illnesses and death. Moreover, climate change could aggravate or increase some allergic diseases [65,89]. This is especially important in cities with Mediterranean climates, where some of the ornamental species native to the Mediterranean and abundant in urban green spaces are among the main allergy-causing agents in the population [90]. A study carried out in the city of Malaga reported that the trend of increasing temperature and atmospheric aridity is the cause of the increasing tendency of *Quercus* pollen production in spring in the western Mediterranean. It concludes that the effect of climate change is mainly reflected in the pollination intensity of anemophilous woody species, which in turn have adapted their flowering time to climate change [91].

In addition, exposure to worse working and living conditions and wider social inequalities will most likely worsen health and wellbeing and increase inequalities in health. For example, forced displacements, housing or job loss, and reduced availability of food and access to it may have an important effect on mental and physical health.

The threat of climate change per se is a key psychological and emotional stressor. People and communities are affected by exposure to information on climate change and its effects. For example, public communication and the media’s messages on climate change and its predicted consequences can affect perceptions of physical and social risks and, as a result, impact mental health and wellbeing [79]. This is especially important among the younger population around the world. A study conducted in 10 very different countries showed that 59% of young people among the countries were very or extremely concerned about climate change and 45% of them said their feelings about climate change negatively affected their daily life and functioning [92]. In the more specific case of the city of Barcelona, according to data from the 2021 Secondary School Risk Factors Survey, 80.4% of girls and 65.4% of boys expressed concern about climate change. Similarly, 86.4% of girls and 78.8% of boys considered that climate change had a negative effect on their lives [93].

### 3.6. Vulnerability

Vulnerability is defined in general terms as the degree to which a person, population, or system can cope with the adverse effects (on health) of climate change, and is determined by the level of exposure, sensitivity and ability to adapt to risk factors [78,94,95,96]. Exposure refers to the degree to which a person or population comes into contact with climate change and its consequences, for example, people working outdoors could be more exposed during episodes of extreme heat. Sensitivity refers to certain individual characteristics that make people more susceptible to being affected by a particular climate-related health problem. For example, the following groups are particularly sensitive to extreme temperatures: older adults, especially those with pre-existing medical conditions; children; women (although some studies attribute their sensitivity to their greater longevity); persons with conditions affecting cognitive functioning, mobility and behaviour; and persons taking medication that can affect sweating or interfere with thermoregulation, such as various psychoactive drugs. Finally, the ability to adapt consists of the presence or absence of key resources and/or adaptive behaviours at the individual or population level including a good water supply, sewage and waste management systems, a transport network that allows access to workplaces and other services, as well as a cool space during prolonged periods of extreme heat; the ability of persons living in energy poverty to have a home with adequate conditions during extreme temperatures (for example, air conditioning, cross ventilation and thermal insulation); and migrants with limited knowledge of the local language or area. Both exposure and the ability to adapt are strongly influenced by the various above-mentioned axes of inequality. Strong social cohesion and support networks among individuals, communities and institutions are essential to foster the ability to adapt among vulnerable populations [11].

### 3.7. Mitigation and Adaptation Policies and Interventions to Climate Change

Climate change prevention measures fall into two main categories: (1) mitigation measures, which aim to stabilize or reduce GHG emissions; and (2) adaptation measures, which aim to prepare for the various effects of climate change, for example, the impact on health. Mitigation and adaptation are complementary approaches to reducing the risks of the impacts of climate change in various time scales. In the short-term, and throughout this century, mitigation could substantially reduce the effects of climate change in the last decades of the XXI century and beyond. The benefits of adaptation can be obtained both by approaching current risks, given that the effects of climate change are already being felt, and, in the future, by approaching emerging risks [97]. Many cities with a Mediterranean climate are already implementing various mitigation and adaptation policies. Some examples of good practices include Cape Town’s water conservation policies, Los Angeles’ heat reduction policies, Seattle’s equity and environment agenda, or Izmir’s climate and health surveillance system.

#### 3.7.1. Mitigation

Reducing GHG emissions requires transforming and making more sustainable the different GHG-generating models mentioned in the framework. While to a large extent such policies require global changes that are likely to take decades, cities can also influence many sectors, which in turn can generate direct co-benefits for health, and more quickly.

In several large cities with a Mediterranean climate, a high impact was observed for air pollution on cardiorespiratory mortality [65]. Promoting policies that advocate more sustainable, safe, and active mobility can reduce GHG emissions and, at the same time, reduce the health effects of air pollution and promote healthy habits such as walking and cycling. Additionally, many Mediterranean cities have a high structural vulnerability to energy poverty [98]. Improving the energy efficiency of buildings can reduce energy poverty and its effects on health and is also a relevant measure to reduce GHG emissions, especially in summer to avoid increased use of air conditioning in these settings. Cities can also bring about changes in the agri-food model, reducing food waste, promoting the consumption of local and eco-friendly food and a lower consumption of meat, or even supporting urban gardens in the streets or on rooftops. Some co-benefits associated with these policies are the reduction of cardiovascular risk associated with a Mediterranean diet rich in local products and with less meat, in additionally greater social cohesion generated in community gardens.

#### 3.7.2. Adaptation

Adaptation policies and interventions can help to reduce the impact of climate change on health. For example, heat wave plans have surveillance systems able to predict future heat waves and are based on alerts that trigger actions to reduce the negative effects on health. Along the same lines, surveillance programs and control of vectors and reservoirs initially detect new vectors and reservoirs and can carry out rapid interventions to minimize the negative effects on the population. Another example includes urban planning measures such as setting up climate shelters to protect the most vulnerable persons during days or periods of extreme heat or increasing urban green space to reduce the heat island effect, especially in the most vulnerable neighbourhoods. It is important to monitor the resilience of social and health services in extreme climate situations to guarantee that they do not pose a risk to patients and that they can cope with events such as power cuts.

Adaptation strategies must not only include, but also prioritize, the most vulnerable groups. For example, heat wave plans should take careful consideration of, among other vulnerable groups, elderly adults living alone, especially women.

Like mitigation policies, most adaptation policies can also have direct and shorter-term co-benefits with respect to the population’s health. For example, the increase in urban green space not only reduces heat exposure and its health impact but can also increase physical activity and social cohesion and reduce atmospheric pollution, factors which also improve the population’s health [11,96,99]. These types of adaptation measures should be encouraged and prioritized [100].

On the other hand, some adaptation policies can have adverse secondary effects. For example, greater use of air conditioning to maintain buildings at a comfortable temperature and the use of desalination to improve the safety of water resources can increase energy demand and therefore GHG emissions.

### 3.8. Climate Justice

Climate justice is a concept that aims to understand the unequal burden of the effects of climate change on health [96,101]. It is, therefore, intimately linked with social inequalities in health and the approach to this problem, since it reveals how the most socioeconomically disadvantaged groups are those with the highest risk of suffering from adverse health effects due to climate change. In this regard, climate justice seeks to ensure that mitigation and adaptation policies generate co-benefits for the most disadvantaged populations with the aim of not widening social inequalities even more [102]. Definitions of climate justice vary, but the concept has three outstanding features: (1) it recognizes the imbalance between responsibilities and harms; (2) it defends the need to implement policies and interventions to correct these imbalances; and (3) it highlights the importance of these policies and interventions to promote human rights, help empower the population and to foster community alliances and self-sufficiency in order to improve the population’s health and wellbeing [103].

Climate justice can be considered on global and local scale: on a global scale, for example, by demanding the discontinued use of fossil fuel and transferring funding from north to south for payment of ecological debt based on historical responsibility [104]; on a local scale, by designing public health policies and interventions to enhance the resilience of the most disadvantaged populations to climate change and its consequences [105]. One example would be to increase the urban green space in the most disadvantaged neighbourhoods, but co-designing, from the outset, measures to avoid green gentrification.

### 3.9. Social and Health Services

Extreme climate phenomena produced by climate change can directly affect social and health services and consequently influence people’s vulnerability and health. Thus, they can hamper access, alter their infrastructure, and cause problems for their functioning. For example, events such as storms, heavy rainfall, and floods can disrupt transport, damage buildings (such as hospitals), interrupt services due to supply cuts, and delay emergency responses.

In addition, the effects of climate change on health (modulated by vulnerability) can lead to changes in the volume and demand for social and health services by, for example, increasing the number of medical emergencies, demand for mental health services and risk of energy poverty, among other effects [89,106,107]. Finally, social and health services work directly to ameliorate the negative health effects of climate change. For example, in Barcelona, during heat waves these services are responsible for, among other things, issuing the advice to be followed by citizens and identifying people whose health is at greatest risk (frail, elderly people, those taking certain medications, homeless people, etc.). These activities should be included in the planning of social and health services to guarantee their resilience.

In contrast, social and health services also play an important role in GHG emissions and consequently mitigation policies. Interventions should also be aimed at these services to reduce their carbon footprint with respect to energy use (heating, air conditioning, water heating, ventilation, etc.), travel (patients, workers, visitors), acquisition of goods and services (medication, equipment, food, etc.), and production of residues. Likewise, adaptation measures are essential to approaching the possible overload on these services that could be produced by climate change, as well as its consequences. Some of these measures improve installations to cope with the increase in the number of persons requiring healthcare, guaranteeing an adequate supply of resources and materials, strengthening mental health services, developing surveillance systems, designing community and support programs for vulnerable people, and providing training programs for persons in these sectors [9,108].

### 3.10. Conclusions and Recommendations

The proposed conceptual framework attempts to explain the relationship between climate change, its causes and consequences and its impact on human health in urban contexts with a Mediterranean climate. The main contribution is the in-depth description of the relationship between climate change and social inequalities in health, attempting to highlight common structural determinants and differential impacts on health by social group. The framework also includes the role of mitigation and adaptation policies and interventions against climate change, highlighting the need to act from the perspective of climate justice and the importance of co-benefits in health that can be provided by such actions. The fight against climate change and its impacts on health must both drive actions, as their combination is essential to provide an immediate response to the public health impacts we are already suffering, as well as avoid or reduce the impacts that are expected in the future.

Climate change already affects many public health functions. This conceptual framework provides a useful contribution to analyse and monitor the effects of climate change on health and health inequalities as well as to guide the design and assessment of policies and interventions to reduce these effects and GHG emissions. The framework reveals the complexity and importance of combatting climate change and its effects on health, highlighting specific particularities of cities with Mediterranean climates. The involvement of multiple sectors, as well as the need for structural changes in models requires intersectoral and multi-level action which, more than ever, requires a health perspective on all policies to improve health equity. Climate change is a complex phenomenon, but the irrefutable scientific evidence, and the already noticeable effects of the current climate crisis, call for strengthening and accelerating the climate strategy both locally and globally.

## Figures and Tables

**Figure 1 ijerph-19-12764-f001:**
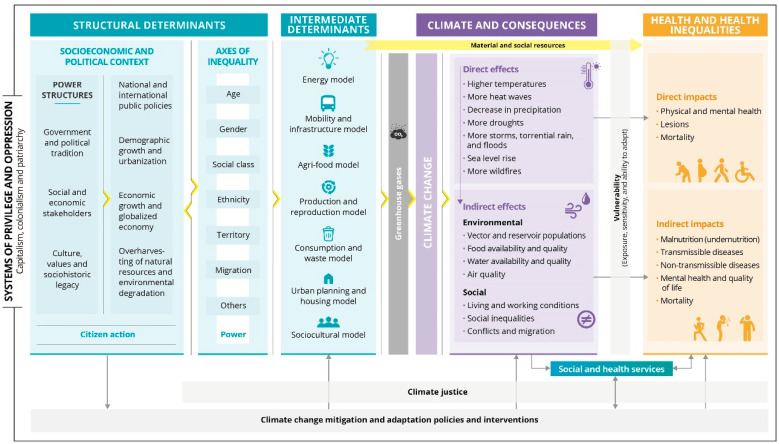
Conceptual framework of the determinants of climate change, its environmental and social consequences, and its effects on health and health inequalities, for urban contexts with a Mediterranean climate and from a social and climate justice approach.

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
