# Peer review of "Climate Change and Health in Urban Areas with a Mediterranean Climate: A Conceptual Framework with a Social and Climate Justice Approach"

_ijerph, 2022, doi:10.3390/ijerph191912764_

Round 1

Reviewer 1 Report

Please see the attached PDF file

Reviewer 3 Report

Dear authors,

This is a very interesting topic, bearing in mind the high risk and important changes that climate change may cause to regions and more specifically to urban areas with a Mediterranean climate. It is a very well-written and strong paper and it was a real pleasure for me to read it. I truly wish to see it published soon. I have only minor comments/ suggestions that you may find below.

1) Keywords: In most journals, it is suggested to use keywords words that do not already appear in the title. If this is also the case for Int. J. Environ. Res. Public Health then you should revise some of your keywords, such as “climate change, Mediterranean climate, health” etc.

2) Line 86: why “increment” and not “increases”?

3) Line 98: “and a high proportion” You could remove this “and” to avoid repetition.

4) Line 105: replace “visitants” with “visitors”

5) Lines 270-271: “and more than 60% of GHG emissions are generated by cities”. This information is already given in line 210.

5) Line 280: reference is needed here.

6) Line 282: what are the GEH-generating models? Do you mean GHG?

7) Line 286: “and education”. Consider replacing it with “ as well as education” to avoid repetition.

8) Lines 316-317: scientific names of species should be written in italics. It also better to give spp. next to the genus. For instance Salmonella spp. instead of Salmonella.

9) Line 320: here you could also add the risk of invasive alien species instead of the more general “the introduction of new ones”

10) Lines 343-345: a reference is may needed here

11) Lines 394-405: I would expect to see some information here about wildfires which are also becoming more and more frequent in urban areas and with numerous human casualties. See for instance the example of Mati in Greece in 2018.

12) Line 413: Salmonella spp. Add a full stop after spp. and use italics for the name of the species.

13) Line 490: avoid using too many “and” in the same sentence

Round 2

Reviewer 2 Report

My comments are fully addressed by the authors and thanks for their efforts. I recommend publishing the paper as it is.